# Indian Parents' Perceptions of Children's Psychological Wellbeing and Academic Learning during COVID-19

Pavneet Kaur Bharaj [1,*], Sarah Hurwitz [2], Nirmal Govindaraju [3], Arya Karumanthra [2], Annie Jacob [4], Sreehari Ravindranath [4] and Adam Maltese [2]

1 School of Social Sciences and Education, California State University Bakersfield, Bakersfield, CA 93311, USA
2 Department of Curriculum and Instruction, Indiana University Bloomington, Bloomington, IN 47405, USA; shurwitz@indiana.edu (S.H.); arkaru@iu.edu (A.K.); amaltese@indiana.edu (A.M.)
3 Eikas Foundation, Chennai 600040, Tamil Nadu, India; nirmal@eikasfoundation.org
4 Dream a Dream, Bengaluru 560011, Karnataka, India; annie@dreamadream.org (A.J.); sreehari@dreamadream.org (S.R.)
* Correspondence: pkbharaj@csub.edu

**Abstract:** Indian children experienced large-scale disruption in educational, psychological, and social welfare access when schools abruptly closed due to COVID-19. In addition to education, the Indian public school system provides services such as meals and benefits related to improving gender parity and indirectly preventing child labor, child marriages, and abuse. Therefore, sustained school closures led to an unfolding disaster in terms of learning loss and multiple unknown effects on children's social and psychological wellbeing. This descriptive study attempts to understand these consequences by asking Indian parents about the emotional, psychological, and academic impacts on their children. Results suggest an adverse impact on children's education and wellbeing. Families reported higher levels of psychological distress, anxiety, and aggression among children. However, the manifestations and ramifications seem to be different—while families from low-income segments struggled to get access to digital devices, others in upper-income segments had to confront excessive device time use. The results suggest that there is a need for a concerted, sustained, multipronged, differential response from the government and civil society to ensure that families can handle these challenges accordingly.

**Keywords:** COVID-19; Indian parents; children's psychological wellbeing; academic learning

## 1. Introduction

The COVID-19 pandemic continues to have a profound impact on the education sector across the globe [1], but ripple effects are particularly evident in India. Indian schools were abruptly closed in March 2020 to reduce the spread of the virus, and many students lost access to services, with deep implications for children's educational, physical, and mental health. India has more than 250 million children in the compulsory school-going ages of 6–14 years [2]—more than six times the size of the same group in the U.S. Around 70% of Indian children live in rural areas [3], hail from low-income communities, and attend public schools [4]. Indian schools were intermittently closed for the majority of two years, interrupting the academic and social support for the largest school-going population in the world. The consequences of this unprecedented disruption are largely unknown.

In India, school support goes beyond education. Public schools ensure that children receive adequate nutrition, help combat child labor and child abuse, serve as a safe space during the day, and have gradually improved gender parity in enrolment, providing girls with more educational opportunities than they had in the past [5]. The sustained school closures, coupled with the economic distress in families [6], caused disruptions to child nutrition [7], interrupted efforts to improve literacy, especially among girls [8] and resulted in dramatic shifts in daily routines.

In addition to losing school as a safe harbor, a rapid transition to remote learning puts families at a crossroads. Online education called for increased use of remote learning technologies, e.g., TV and virtual meeting platforms [9]. With no preparation or infrastructure, parents needed to take on more responsibility for their children's education. Parents, teachers, and schools felt unprepared and overwhelmed by remote learning [10] as they navigated personal, technical, logistical, and financial barriers [11]. For students in rural areas, online learning was often unavailable when the home lacked computers or tablets and electricity or Wi-Fi was in limited supply.

Throughout the pandemic, educational stakeholders contemplated the consequences of extensive school closures on the academic, social, and emotional wellbeing of students. This paper attempts to assess the effect of sustained school closures on the wellbeing of Indian children (aged 6–14 years) as shared by their parents/caretakers. We hypothesized that school closures had a significant impact on children's learning and behavioral patterns. The primary research question we sought to answer is: *How did the COVID-19 pandemic affect students' psychological wellbeing and academic learning, as reported by their parents/caretakers?*

## 2. Materials and Methods

We used purposeful sampling to survey parents and caretakers across multiple socioeconomic segments. A few of our research team members were closely working with Indian families to provide educational opportunities to their children; hence, they had a clear idea of the characteristics or attributes they were interested in studying. We wanted to capture the voices of parents/caretakers in our survey explicitly while answering the proposed research question. We used a survey design to gather information from our participants. Using a survey allowed access to a geographically disparate range of families within India. This study was approved by the Indiana University Bloomington's Institutional Review Board (IRB #10098).

### 2.1. Survey Development

We aimed to gather information on the effect of sustained school closures during COVID-19 on student learning and mental health from the perspective of parents/caretakers. Questions were developed via a partnership between researchers from Indiana University Bloomington in the United States and a non-governmental, non-profit educational foundation in India (Eikas Foundation). Broadly, items were written to learn about children's access to school and other learning opportunities, changes observed by parents in their child's behavior, and parents'/caretakers' concerns about their child's wellbeing during the height of the pandemic. To minimize the complexity of answering items and analytic interpretation, participants (parents/caretakers) were instructed to focus their responses on the experience of their child between the ages of 6–14 years (which is the compulsory school-going age in India). A copy of the survey can be found in Supplementary Materials.

The research team initiated the first phase of survey development in early 2021. The initial versions were piloted by Author 3, leading the partnered Foundation with a small group of individuals from Chennai, a southern state in India, during July and August 2021. Ten respondents participated in this phase, seven via in-person interview and three via phone calls. We deliberately refrained from utilizing recording devices during interviews to foster an environment of trust and security among the participants. The presence of cameras or audio recorders might have induced apprehension in our interviewees, thereby potentially impeding the candid sharing of their information. Instead, we relied on the discerning memory and expertise of the interviewer, who maintained close ties with the community under study. After each interview, the interviewer compiled field notes, which subsequently formed an integral component of our research findings. This methodology aligns with a practice that has evolved over time in certain social science research studies [12,13]. We used this data to refine the survey questions and prompted the inclusion of new questions.

Next, the survey questions were finalized and uploaded to Qualtrics, an online platform (Qualtrics, Provo, UT, USA). Qualtrics allows for mobile and computer-based responses and was accessible for respondents across locations in India. This also helped in facilitating mobile accessibility since many Indians' primary internet access is via their cell phones. The survey included a set of socio-demographic questions, including occupation, education, and area of residence. A set of questions about school closures included how often the student attended school during the pandemic, whether school was held in-person, online, or if school was fully closed, and how caretakers supported their child's remote learning. Items asked about barriers to remote education (e.g., access to a computer or tablet; availability of internet). Finally, questions captured parents'/caretakers' perceptions about the behavioral and lifestyle activities of their child (e.g., were they able to play outside or socialize with others; did they spend more/less time on electronic devices). Response options were Likert-type scales or open-ended. Open-ended responses were captured for questions like how much time the children spent on different activities (e.g., studying, playing, doing chores). At the end of the questionnaire, respondents had the opportunity to add general reflections on remote learning.

The online version of the survey was publicly disseminated using social media platforms, such as WhatsApp, Facebook, and LinkedIn, from late November 2021 onwards, and we continued to seek responses until January 2022. Information about the purpose and conduct of the research was provided with the post. Parents completed the survey in two ways: either via digital self-report (i.e., the online/mobile Qualtrics form) or by having an interviewer read the survey items aloud and record responses on the respondent's behalf. The interviews were offered to families to reduce digital access (no access to social media platforms) and language barriers common in India. We approached a large, well-established educational non-profit organization in India (Dream a Dream; Author 5–6) to assist with the one-on-one data collection. Their research team collected survey data in-person, using a paper–pencil form of the questions that had been translated into Kannada, the vernacular of Bengaluru. Six field resource facilitators administered the survey to 70 parents in low-income settings in the Fall of 2021. The interviewers read the survey questions to the parents and wrote down their answers. These responses were then imported into Qualtrics.

*2.2. Participants*

Our data include surveys collected from 165 parents/caretakers with at least one child aged 6–14 years. Table 1 provides the socio-demographic information of the respondents. Most surveys were completed online (58%), and the rest were collected by the partnered Foundation's interviewers. Many online respondents were from Punjab in India's north (36%) and Tamil Nadu, Telangana, Karnataka, and Andhra Pradesh in the south (53%). Most respondents were between 25–40 years (57%) and approximately 27% of respondents were K-12 graduates. About 63% were in full-time or part-time employment, with 22% identifying as homemakers. Most of the students in our sample attended private schools (73%). At the time of surveying, half of the parents reported their child was attending some in-person school (53%), with one-third (32%) reporting online schooling. Mobile phones were the most common technology used to access lessons (53%), followed by computers (27%).

*2.3. Data Analysis*

We used two distinct approaches to analyze the data. We used descriptive analysis to understand the trends captured in the responses from open-ended survey items. In addition, we analyzed information from the initial set of ten interviews. Author 1 reviewed these interview notes line-by-line to identify common themes [14]. The phrases were paired based on a common meaning using in vivo codes, which allowed us to keep the interpretations close to the participants' originally expressed meanings. Since these interviews were not recorded and the interviewer (Author 3) captured field notes, verbatim answers are not

available. Where relevant, quotes are included from open-ended textboxes or notes taken by the interviewers.

**Table 1.** Socio-Demographic Information of Respondents (*N* = 165).

|  | Percentage (%) |
|---|---|
| Age (in Years) | |
| Under 25 | 1 |
| 25–40 | 57 |
| 40–60 | 29 |
| No response | 13 |
| Educational attainment | |
| Elementary (Grade K–8) | 5 |
| Secondary (Grade 9–12) | 22 |
| Undergraduate | 18 |
| Graduate | 32 |
| Illiterate (No Education) | 1 |
| No response | 23 |
| Profession | |
| Housewife | 22 |
| Private Job | 18 |
| Business | 8 |
| Unidentified | 9 |
| Teaching | 29 |
| No response | 15 |
| School Type | |
| Public | 9 |
| Private | 73 |
| Aided | 9 |
| No response | 9 |
| Modality of School (when data was gathered following Pandemic) | |
| In-person | 53 |
| Online | 32 |
| Hybrid (both in-person and online) | 1 |
| Not attending school | 6 |
| No response | 8 |
| Pandemic Schooling Modality | |
| Computer | 27 |
| Mobile | 53 |
| Tablet | 6 |
| No response | 15 |

## 3. Results

### 3.1. Children's Psychological Wellbeing during COVID-19

When asked about their child's wellbeing, most parents reported being more concerned than they were before the pandemic, especially regarding their child's health, education, and safety (Figure 1). Many participants reported that disruptions in routine resulted in greater consumption of screen time, which prompted worries about their children not getting enough time for exercise and play. The interview data underscored that online learning and spending more time at home resulted in excessive use of digital devices, often mobile phones. An accountant whose son attends a private school explained, "Now the child is spending 8–10 h on screen time (apart from online school screen time), which has led to a lower concentration on studies" (Interview 6). His son complained of eye pain, for which the doctor recommended curbing screen time. Similarly, a pediatrician reported

that her 10-year-old became addicted to mobile photography and spent hours each day capturing and editing pictures (Interview 2).

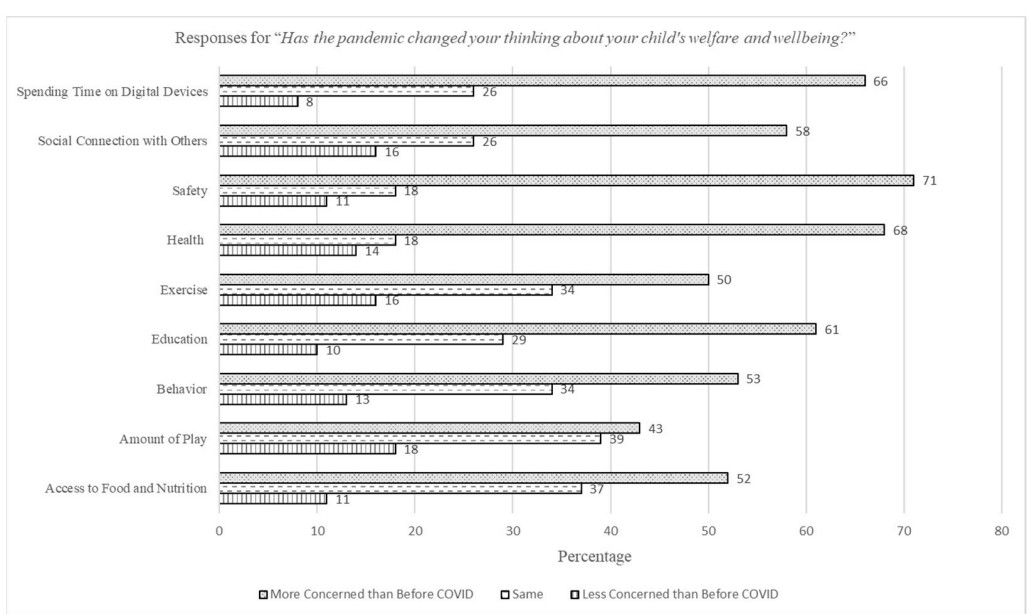

**Figure 1.** Responses for *"Has the pandemic changed your thinking about your child's welfare and wellbeing"* by percent.

On balance, a housekeeper shared that his daughter was able to use a phone for her studies for only half an hour as the device was shared by older siblings for their matriculation examination (Interview 1). This response suggested that the use of digital devices is related to the aspect of accessibility of the families—with some needing to share a device among multiple family members.

In addition, school closures meant that children were not provided with school meals or even safe spaces to play and exercise, which concerned many families. Around half of the respondents expressed that they were more concerned about their child's exercise routine than before COVID-19. In the qualitative interviews, one father expressed concern about the health of his son, who had put on a lot of weight. He explained that his son used to play outside for more than three hours every day before COVID-19, but during lockdown and due to social distancing, such playtime was significantly reduced (Interview 10). He further expressed worry about the child's mental health as he felt that his son might be feeling lonely, stressed, or fed up during COVID-19 days.

Nearly 60% of parents were more concerned about how the pandemic impacted their child's social connection with others (Figure 1). Students did not have access to social connections with their peers or teachers, which might have impacted their socializing skills. When asked if the parents/caretakers have noticed any psychological changes in their child during the pandemic, most parents reported that their children had more stress, anxiety, and aggression during the pandemic, both for themselves and for their families (Figure 2). For a follow-up prompt, did any of your children express concerns to you during the pandemic? If so, what was the nature of these?, respondents expressed worries that children heard news reports and overheard discussions which provoked uneasiness. "Yes, when he used to watch the news or when his friends discuss regarding the family deaths—he gets anxiety and fearful", as shared by a mother of three children who is employed as a design engineer. This resulted in fears at home and in the community. "My daughter is worried about my [parents'] health safety," according to an account shared by a mother of two children from South India, who also runs a private tailoring business, while a father of one child, employed as a supervisor in a garment store reported, "My child is afraid to go outside and does not want to meet anyone." This indicates that children expressed concern

about the wellbeing of their loved ones during COVID-19, which either made them more fearful about the situation or prepared them to take lead roles in the family; for example, starting to instruct their families regarding COVID-19 protocols. Parents also reported that children were anxious about the family's finances because parents lost jobs or were forced to work less due to lockdowns.

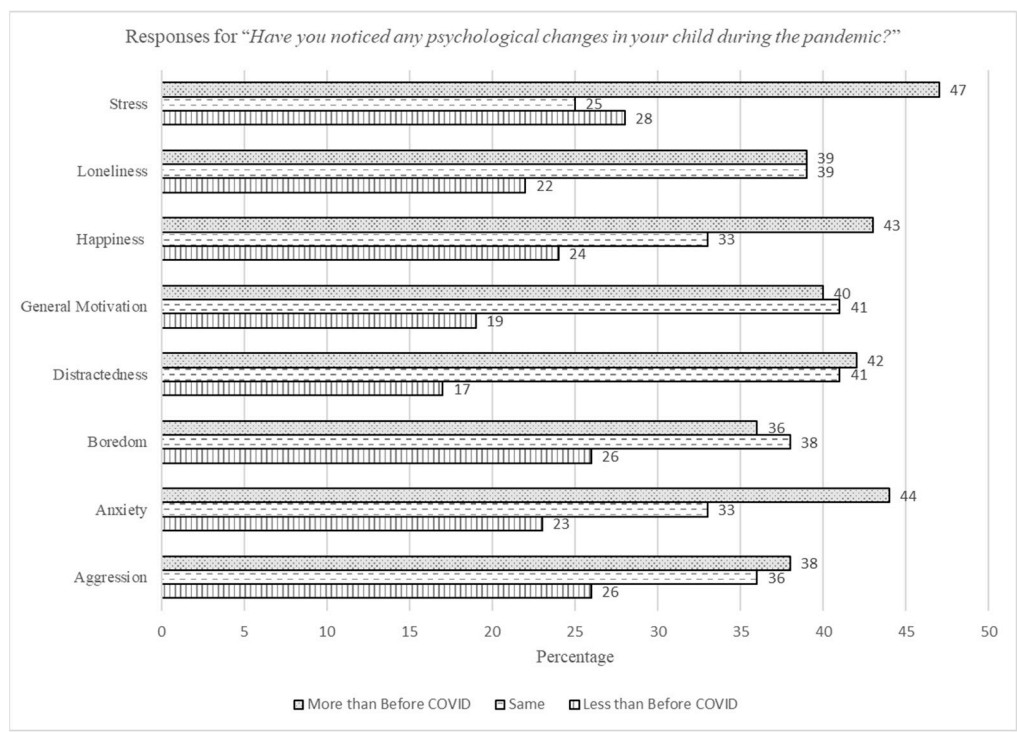

**Figure 2.** Responses for "*Have you noticed any psychological changes in your child during the pandemic?*" by percent.

More than 40% of respondents shared that their children felt happy as they were not required to go to school. Children's happiness was related to not going to school or getting a chance to browse the internet or watch TV whenever they wanted (Interviews 5 and 6).

### 3.2. Children's Academic Growth during COVID-19

To understand how the pandemic changed children's academic growth, parents were asked, "In your judgment, how much learning or progress did your child make in the past year?". Respondents were split, with some reporting that their kids either lost academic skills or made no academic progress (27%), while others reported that their children made a tiny bit to a great deal of academic progress (Figure 3). Compared to in-person schooling, 36% reported learning being worse than before COVID-19; however, 24% of respondents selected better than before COVID-19.

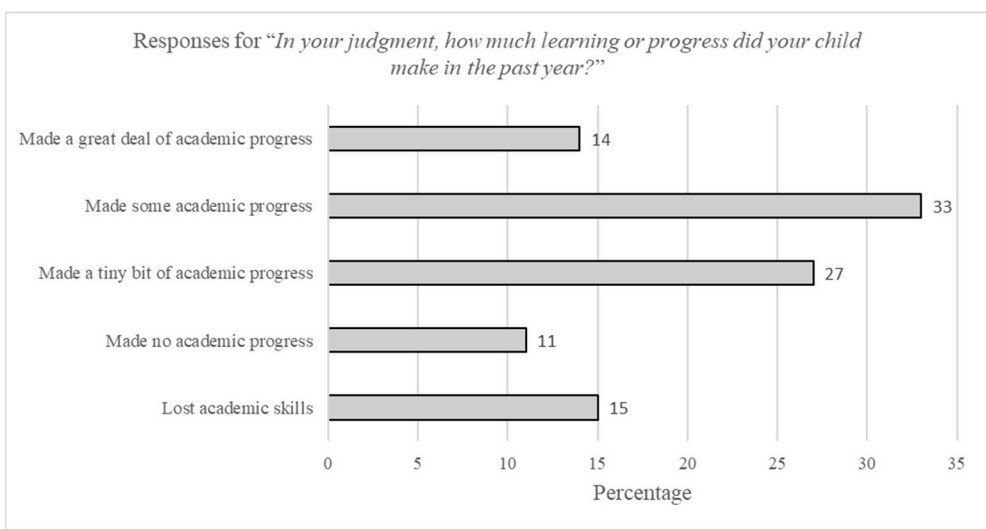

**Figure 3.** Percentage of respondents reporting children's learning progress.

## 4. Discussion

In this descriptive study, we sought to examine the effects of sustained school closures on the wellbeing, psychological changes, and academic learning of children in India, as reported by their parents/caretakers. Nearly two-thirds of respondents expressed concern about their children spending excessive time on digital devices. Since many schools went online, virtual classes made it hard for parents to control screen time for those who had mobile devices or computers [15]. The cancellation of in-person activities, less outdoor play, and the accessibility of devices as entertainment contributed to excessive screen exposure [16].

Furthermore, screen time behaviors differed based on the families coming from different income groups [17]. Although we do not have definitive data to accurately characterize the financial situation of each family, we were able to make some general comparisons using the information about participants' self-reported occupations. When adding in this layer of information, the results suggest that parents from low-income segments may not have shared concerns related to screen time due to lack of access to devices. In comparison, middle-income groups were more likely to share that their children got addicted to devices—even to the extent that required medical attention. The dangers related to excessive screen time hampering learning capacities in children as well as causing psychological problems, including a greater risk for depression or anxiety [18] and physical disorders like disrupted sleep patterns, obesity, dry eyes, and hypertension, are well documented [10,19].

In relation to physical health, the lack of access to parks and public spaces has impacted children's exercise and physical routines. In addition, parents also restricted children's use of public places due to transmission concerns [20]. Researchers cautioned that the lockdowns and school closures might disrupt everyday routines and would come at the cost of an existing public health challenge, i.e., childhood obesity [21]. We believe that lower physical activity may also be attributed to low levels of independent mobility (i.e., the freedom to travel around the neighborhood without parental supervision).

The evolving pandemic brought new educational needs. Remote learning is dependent upon factors like access to digital devices and internet connectivity. Though India is one of the world's second-biggest smartphone markets, for millions of children, especially those from lower socioeconomic backgrounds, inequalities persist [22]. About 29.6 million children in India did not have any access to a digital device in 2021 [23], and almost 70% of the survey respondents in our sample stated lack of internet as one of the key challenges during online schooling, which aligns with prior studies [24].

Countries like India were struggling with the learning crises even before the pandemic [9]. COVID-19 exacerbated these issues, and it seems likely that the heaviest toll will

manifest on traditionally marginalized and under-resourced communities. The impacts of COVID-19 on learning will take years to assess, but initial research is rather bleak and shows significant losses for Indian children [22]. In our sample, approximately 25% of parents reported that their kids lost academic skills or made no academic progress during the pandemic. Supporting this, [25] reported that "more than two-thirds of parents of students who received at least some online instructions are concerned about their children falling behind in school" (p.n.d.). The sudden transition to virtual instruction required families to support their children academically. Parents were unequipped to support their children's educational needs and looked for avenues to meet this requirement. For instance, they watched videos to learn mathematical concepts or created family scenarios where elder siblings played supportive roles. One participant—an accountant with a son attending private school—expressed dissent towards online learning. He shared how unmonitored online learning negatively impacted the academic progress of his son, especially with the assessment situations. This observation resonated with our prior work, suggesting that online instruction demands parents to remain proactive to ensure that children learn the correct information [26].

Contrary to this, we were surprised to find that 14% of parents reported that their kids made a great deal of academic progress during COVID-19. Digging into the data helps us to interpret these differences. Approximately 25% of our respondents belonged to teaching professions or were homemakers, which might have given them an edge in taking care of their child's educational needs. Also, the families with fewer financial struggles seemed more able to contribute during the remote instruction. As an example, a pediatrician from South India (Interview 2) reported that her son ". . . started reading a lot because he had a lot of free time" (because of the shorter duration of online classes). This parent shared that basic reading and writing skills need attention, and paying attention to these foundational structures helped her son with his academics. As a medical professional with financial stability, this mother's circumstances allowed her to invest unusual time and effort, which might not be possible for an average middle or low-income family. Mak (2021) also noted that Indian children coming from richer households were motivated to learn during the pandemic, as "monetary resources might play a more influential role in shaping a child's motivation during the pandemic than whether a parent has more time to supervise their children" (p. 4).

In India, a huge proportion of the population comes from lower-income groups. The pandemic disrupted the flow of income in households due to unemployment and reduced work opportunities, which further led to concerns related to access to food and nutrition. Closure of schools exacerbated food insecurity and raised alarms for long-term health impacts on economically disadvantaged families [7]. Despite the Indian Prime Minister's Overarching Scheme for Holistic Nourishment Programme (POSHAN Abhiyan), 1.2 million children were reported to be moderately and severely malnourished by April 2020, and this number increased to 3.3 million by October 2021 [23]. Exacerbated by existing inequities, these factors will have long-term implications for nutrition and health, such as inadequate dietary intake and disease, finally leading to all forms of malnutrition. The impacts of this on youth—likely negative in terms of wellbeing and learning—will take years to understand.

When it comes to psychological impact, parents expressed concerns related to the emergence of negative emotions (stress, anxiety, aggression, loneliness, and boredom) in children. Approximately half of the respondents reported more concerns about their children's stress levels, and 40% expressed concerns related to anxiety and aggression. Parents also reported increased signs of aggression and anger (Interviews 1, 5, 6, 7). One father (Interview 5) shared that his daughter started beating her younger brother without any reason, which was not the case when schools were open. Authors [23] asserted that anxiety, depression, irritability, and inattention are the predominant psychological symptoms observed in Indian children during COVID-19. Researchers [10] cautioned that the life disruptions caused by COVID-19 have the potential to leave lasting psychological

impacts on individuals in affected communities. These concerns are also not unique to Indian parents, as a national study of American parents with children aged under 18 years found that 14% of parents reported a decline in behavioral health for their children since the pandemic began [27]. In addition, American parents have also reported a negative impact on their children's sleep quality [28] and eating habits [29].

There can be multiple reasons for the increased stress in children, like fear of contracting or losing their loved ones to COVID-19 or even social isolation due to school closure [10]. In our sample, parents/caretakers expressed that staying at home without having enough access to the outside world led to feelings of boredom, restricted social interactions, and decreased quality of education (Interview 1, 2, 9, 10). One of the respondents even stated that kids were feeling as if they were in jail, leading to multiple instances of crying and aggressive behavior (Interview 1). One thing to note is that a close-knit family structure allows some families to utilize support from their loved ones. Interviews revealed that access to grandparents, cousins, and uncles/aunts helped in handling the pandemic-related stress (Interviews 6, 9).

The vicious convergence of pandemic, lockdowns, loss of jobs, and financial conditions negatively influenced the type of resources children were receiving from their parents. Parents shared these implicitly in their interviews as they felt helpless to assist their children due to financial burdens. Our qualitative interviews revealed that a parent working as household help was unable to send his daughter to extra tuition or online classes due to a lack of monetary resources (Interview 4). There was a transition in his daughter's life at both academic and personal levels because, before COVID-19, the daughter was never asked to work on household chores as the family was of the opinion that the education of the daughter might help them gain dignity in society; hence, wanted to support her education with all resources. However, during COVID-19, the mother started to teach the daughter about household chores because the family started to get concerned about their daughter's educational competency due to a lack of resources. One auto driver (Interview 5) shared how his financial situation was impacted by the lockdown. He shared that before the lockdown, he was able to take home ~₹800 ($11) per day, but during the pandemic, he was not earning anything. Before the lockdown, he bought snacks (~₹200 ($3)) every day for his kids, but he could not manage that without income. In a few cases, it reflects stories of multiple Indian families where children bore a negative impact on their psychological wellbeing due to lost employment options [30].

## 5. Conclusions

Despite interesting results, there are limitations to this study. We distributed our survey through multiple channels; however, most respondents came from two Indian states—Tamil Nadu and Punjab. Further, our data came from only 165 survey responses and 10 early-stage interviews. Alone, these data cannot lead to generalizable inferences. In addition, we used purposeful sampling due to the nature of this study; however, we acknowledge that the presence of a non-randomly selected sample might have inevitably influenced the outcomes, and this must be considered when interpreting our results. In this study, we gathered socio-demographic information related to several key factors, including age, gender, education, marital status, and geographic location. While we recognize that income can be an essential factor in understanding socioeconomic disparities and their potential impact on the outcomes studied, our analysis is limited to the available socio-demographic variables. We would encourage researchers to explore the interplay between socioeconomic factors and the academic and psychological experiences of students during and after the COVID-19 pandemic in their future research. Though there are a few limitations, our results generally align with emergent research from others on the impacts of the pandemic.

The COVID-19 pandemic brought a complex array of challenges, with many having negative repercussions on children's wellbeing. These findings are in tune with what educators across the globe expressed about students' unusual social and emotional con-

ditions during the pandemic. While the COVID-19 pandemic is no longer at its peak, its repercussions are still very much felt, and the lessons learned during this period remain relevant. While we may be moving beyond the acute phase of the pandemic, its consequences will continue to shape educational and social landscapes for years to come. We expect that our findings may be useful to educators and decision-makers to learn about the experiences of Indian parents/caretakers and recognize how the pandemic has left lasting impacts, especially on children's wellbeing and learning. The disruptions caused by the pandemic have had a profound and sustained effect on students, and understanding these effects is critical for informed decision-making and policy formulation. Knowledge of these experiences can provide insight into where efforts should be focused to address some of the educational, social, and emotional challenges being faced by Indian families. Issues related to malnutrition, obesity, or poor health increased at a faster pace due to lifestyle changes during the pandemic, for example, a lack of physical activity, increased screen time, or disturbed sleep patterns. Any interventions geared towards safely supporting children's wellbeing must take into consideration the individualized needs of the households to ensure that the support reaches vulnerable communities.

The pandemic posed more than health and learning challenges to society, including triggering a lot of psychological and mental issues. Researchers [31] found that a higher level of stress, depression, and anxiety is positively correlated with each other and negatively with wellbeing. There are long-term consequences of such feelings, not only on their mental health but also on academic growth [32]. Our study's insights are relevant for designing long-term strategies that can help students recover from the impact of the pandemic on their psychological wellbeing and academic growth. We believe that the impact of such negative feelings can be mitigated using effective emotional support, which can come from professionals (e.g., psychologists' services [26]) or even family members and friends [10]). Such support systems might proactively prevent psychosocial crises as well as foster wellness.

We feel that our findings are not confined to the COVID-19 period alone; they hold valuable insights for the ongoing recovery and future preparedness. Moving forward, we need to consider "schools as places of work, places of care, and places of learning—because they are simultaneously all three—'normal' is not a standard to which we should aspire" [33] (p. 36). To ensure that schools do not deepen the existing learning losses, we need to mitigate these amplified educational inequities with the concerted efforts of school officials, families, educators, and policymakers alike [26,27]. Such policies need to be crafted to ensure that families without enough financial access have the provision of resources to support online learning. The study advocates for the appropriate allocation of resources to provide maximal support for vulnerable populations during periods of disruption requiring remote learning, as well as the allocation of mental health services to support parents through these changes. The informed provision of resources and support, particularly among those most vulnerable, might help in mitigating the adverse impacts of the pandemic upon families.

**Supplementary Materials:** The following supporting information can be downloaded at: https://www.mdpi.com/article/10.3390/educsci13111146/s1.

**Author Contributions:** Conceptualization, P.K.B., S.H., N.G., A.J. and A.M.; Methodology, P.K.B., S.H., S.R. and A.M.; Software, P.K.B. and A.M.; Validation, S.H.; Formal analysis, P.K.B. and A.M.; Investigation, N.G.; Resources, N.G., A.J. and S.R.; Data curation, P.K.B., A.K. and A.M.; Writing—original draft, P.K.B.; Writing—review and editing, P.K.B., S.H., N.G., A.K., A.J., S.R. and A.M.; Visualization, P.K.B. and A.K.; Supervision, N.G. and A.J.; Project administration, N.G., A.J. and S.R. All authors have read and agreed to the published version of the manuscript.

**Funding:** This research received no external funding.

**Institutional Review Board Statement:** The study was conducted in accordance with The Indiana University HRPP and approved by the Institutional Review Board of Indiana University Bloomington.

**Informed Consent Statement:** Informed consent was obtained from all subjects involved in the study.

**Data Availability Statement:** The data supporting reported results can be found by contacting the first author at pkbharaj@csub.edu.

**Acknowledgments:** We thank Ravichandra K. from Dream a Dream for his expertise and assistance during the data collection phase of this study.

**Conflicts of Interest:** The authors declare no conflict of interest.

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
