# Peer review of "Indian Parents’ Perceptions of Children’s Psychological Wellbeing and Academic Learning during COVID-19"

_education, doi:10.3390/educsci13111146_

Round 1

Reviewer 1 Report

Comments and Suggestions for Authors

The article entitled “Indian Parents’ Perceptions of Children’s Psychological Wellbeing and Academic Learning During COVID 19” consists of an interesting study about the social and psychological well-being of children During COVID 19.

Although this is an important study, it needs further improvement. Below are some parts of this article theauthors need to improve.

Although the writing quality is very good. There are a few sentences that need improvement and errors that need to be corrected. For example, on page 2, the authors wrote

A few of our research team members were closely working with Indian families to provide educational opportunities to their children, hence, they had a clear idea of the characteristics or attributes they were interested in studying.” There should be a semicolon before hence, not a comma.

On the same page, the authors wrote, “Next, the survey questions were finalized and uploaded to an online platform, Qualtrics (Qualtrics, Provo, UT).”

This sentence can be improved by writing, “Next, the survey questions were finalized and uploaded to Qualtrics, an online platform (Qualtrics, Provo, UT).

On page 4, the authors wrote, “Since these interviews were not recorded and interviewer (Author 3) captured field notes, verbatim answers are not available.”

It is very unusual to conduct interviews and not use verbatim responses when reporting the results. Just because the interviews were not recorded does not mean that verbatim responses cannot be provided.

The authors should read “Conducting in-depth interviews with and without voice recorders: a comparative analysis” (available on the internet) to see that accurate verbatim responses can be provided without using voice recorders.

The authors state on page 2 that they “used survey design to gather information from our participants.” And they provide information on how participants responded to various questions. However, this is not enough, they need to include a figure that lists all the questions on the survey.

Comments on the Quality of English Language

Although the writing quality is very good. There are a few sentences that need improvement and errors that need to be corrected. For example, on page 2, the authors wrote

Author Response

Your feedback is greatly valued. We sincerely appreciate the careful review and thoughtful considerations you've provided for this manuscript. Kindly peruse the changes we've made in response to your concerns

  • Although the writing quality is very good. There are a few sentences that need improvement and errors that need to be corrected.

For example, on page 2, the authors wrote “A few of our research team members were closely working with Indian families to provide educational opportunities to their children, hence, they had a clear idea of the characteristics or attributes they were interested in studying.” There should be a semicolon before hence, not a comma. --> we have fixed this. We also found a similar concern on Page 9, we have also handled that.

On the same page, the authors wrote, “Next, the survey questions were finalized and uploaded to an online platform, Qualtrics (Qualtrics, Provo, UT).” This sentence can be improved by writing, “Next, the survey questions were finalized and uploaded to Qualtrics, an online platform (Qualtrics, Provo, UT). --> we accepted this change

  • On page 4, the authors wrote, “Since these interviews were not recorded and interviewer (Author 3) captured field notes, verbatim answers are not available.” It is very unusual to conduct interviews and not use verbatim responses when reporting the results. Just because the interviews were not recorded does not mean that verbatim responses cannot be provided. The authors should read “Conducting in-depth interviews with and without voice recorders: a comparative analysis” (available on the internet) to see that accurate verbatim responses can be provided without using voice recorders. --> We extend our sincere gratitude to the reviewers for generously providing us with this valuable resource. Prior to their feedback, we were unaware of the readings mentioned. As a result, we have incorporated a wealth of guidance and implemented revisions on page 2 of the manuscript. On this page, we have clarified that while we may not possess verbatim transcripts, we do have comprehensive field notes derived from interviews.

  • The authors state on page 2 that they “used survey design to gather information from our participants.” And they provide information on how participants responded to various questions. However, this is not enough, they need to include a figure that lists all the questions on the survey. --> Thanks for your suggestion. We've included the entire survey as supplementary materials for the readers' reference.

Reviewer 2 Report

Comments and Suggestions for Authors

Thank you for inviting me to review this paper. The writing of this paper is good. However, there are two major concerns. First, the most important one, the quantitative method or more precisely to say, the descriptive statistics conducted by the authors, can not serve the research question well. The research question is: “What impacts did the COVID-19 pandemic have on students’…” I was expecting that the authors would conduct experiment designs or quasi-experimental designs to obtain causal inference since they use “impact” in their research questions. However, only descriptive statistics were conducted. Second, since we are entering the post-COVID-19 era, the research and practical implications of this study which investigates the COVID-19 period should be better illustrated.    

Minor issue:

1. When the survey was conducted, this should be pointed out.

2. For the qualitative part, participants should be coded. For example, in line 160, this father could be coded as F1. Also, the quote from the interview could be italicized.

Author Response

We want to express our heartfelt appreciation for the thorough and insightful feedback. Your constructive comments have been instrumental in improving the quality of our work, and we truly value the time and effort you have dedicated to our manuscript. 

  • First, the most important one, the quantitative method or more precisely to say, the descriptive statistics conducted by the authors, can not serve the research question well. The research question is: “What impacts did the COVID-19 pandemic have on students’…” I was expecting that the authors would conduct experiment designs or quasi-experimental designs to obtain causal inference since they use “impact” in their research questions. However, only descriptive statistics were conducted. -->This was a great observation that had escaped our initial attention. We wholeheartedly concur with your insight regarding the potential causality implications of the word "impact." As a result, we have revised the research question to read as follows: "How did the COVID-19 pandemic influence students' psychological well-being and academic learning, as reported by their parents/caretakers?" This revised question aligns more accurately with the nature of our study, as we did not employ experimental designs but collected data at a single time point. We hope this addresses your concerns.

  • Second, since we are entering the post-COVID-19 era, the research and practical implications of this study which investigates the COVID-19 period should be better illustrated.    --> We are grateful for how you prompted us to reevaluate the significance of our study in both current and future contexts. We have provided a detailed explanation of this in the conclusion section, spanning pages 9 and 10 of the manuscript. We trust that this adequately addresses your concerns.

Minor issue:

  1. When the survey was conducted, this should be pointed out. --> Thanks for pointing this out. We clarified the timeline of the data collection for our readers on p. 2 and 3 of the manuscript.
  2. For the qualitative part, participants should be coded. For example, in line 160, this father could be coded as F1. Also, the quote from the interview could be italicized. -->As our pool of interviews was quite limited, we identified them using labels like Interview 1, Interview 2, and so forth. We have incorporated your suggestion to italicize the interview quotes, a change that has been consistently implemented throughout the manuscript.

Reviewer 3 Report

Comments and Suggestions for Authors

Excelent reserach report. Better graphics of perhaps haotrizantlly alligned, mya inclrese visual presentation of tdata. 

Author Response

Thanks a lot for your feedback. We have added the horizontally aligned graphics in the manuscript. 

Reviewer 4 Report

Comments and Suggestions for Authors

The authors adequately recognize the different positions and assumptions regarding the covid pandemic and the effects that this could have on the emotional well-being of students, particularly in India. The objective of the work is to verify if there are effects on the well-being of students associated with this pandemic and the hypothesis they put forward is that there will indeed be effects as a consequence of this pandemic. It would be interesting to guide or continue the study or the description of the results based on these two initial categories that they are proposing in the discussion they take up the differences attributable to the socioeconomic level but they are not contemplated within the part of the results and their analysis, the type of study is not mentioned, the ethical aspects of the research are mentioned and also the limitations that the work has as a consequence of having a sample that is not randomly selected.

Author Response

We want to express our sincere gratitude for your valuable feedback. Your insights and suggestions are highly appreciated as they contribute significantly to our continuous improvement. Thank you for taking the time to share your thoughts with us. We have addressed your concerns in the manuscript as follows:

  1. The objective of the work is to verify if there are effects on the well-being of students associated with this pandemic and the hypothesis they put forward is that there will indeed be effects as a consequence of this pandemic. It would be interesting to guide or continue the study or the description of the results based on these two initial categories that they are proposing in the discussion they take up the differences attributable to the socioeconomic level but they are not contemplated within the part of the results and their analysis --> This is such a great suggestion. Thanks for sharing; however, we gathered socio-demographic information related to several key factors, including age, gender, education, marital status, and geographic location. However, we did not collect data on income. Our study focuses on the data that we did collect, and we have analyzed the socio-demographic factors at our disposal to the best of our ability. The absence of income data does not negate the value of our findings within the scope of our research objectives. We believe our study still contributes meaningful insights, but we acknowledge the importance of considering income in future investigations to obtain a more nuanced understanding of socioeconomic factors. (A) We have added this as the limitations section of the paper. (B) We have also made necessary changes in the text of the manuscript so that the readers do not infer that we are making claims across socio-economic level of Indian families.

2. the type of study is not mentioned --> Thanks for bringing this to our attention. We have clarified for the readers that this is descriptive study in the abstract as well as in the body of the manuscript.

3. the ethical aspects of the research are mentioned --> We have mentioned in Section 2 of manuscript "This study was approved by the university Institutional Review Board (IRB #10098)."

4. also the limitations that the work has as a consequence of having a sample that is not randomly selected. --> Thanks for sharing this with us. We have made revisions in the limitations section and stated that the presence of a non-randomly selected sample might have inevitably influenced the outcomes, and this must be considered when interpreting our results

Round 2

Reviewer 1 Report

Comments and Suggestions for Authors

The authors did a good job in revising this paper; however, a few minor revisions are still needed. On page 6, the authors included the following verbatim quote:

“Yes, when he used to watch the news and friends discuss regarding the family deaths, he gets anxiety and fearful.”

The authors did not indicate who provided this quote like they did with the other quotes. 

It also appears that this quote was translated into English incorrectly. It appears that the correct translation is the following:

 “Yes, when he used to watch the news and friends discuss the family deaths, he experienced anxiety and fear.”

On page 9, the authors started a sentence with “Not only this,” These are awkward words to use to start a sentence. “In addition,” would be an example of better words to use here.

Comments on the Quality of English Language

The quality of English is good.

Author Response

We greatly appreciate the thorough consideration you've given to our manuscript. We have incorporated the desired changes in this revised version. This is how we have addressed your concerns. 

The authors did a good job in revising this paper; however, a few minor revisions are still needed. On page 6, the authors included the following verbatim quote: “Yes, when he used to watch the news and friends discuss regarding the family deaths, he gets anxiety and fearful.” The authors did not indicate who provided this quote like they did with the other quotes. --> We have added this information in the revised document (line 200-206). 

It also appears that this quote was translated into English incorrectly. It appears that the correct translation is the following:  “Yes, when he used to watch the news and friends discuss the family deaths, he experienced anxiety and fear.” --> we have made revision for the readers. please refer to line 200-201 of the document.

On page 9, the authors started a sentence with “Not only this,” These are awkward words to use to start a sentence. “In addition,” would be an example of better words to use here. --> we have made this revision by replacing "not only this" with "in addition"

Reviewer 2 Report

Comments and Suggestions for Authors

Thank you for the revised manuscript. The manuscript was improved, and I have no further comments.

Author Response

We would like to express our profound gratitude for your insightful and constructive feedback provided during the initial review process. Your valuable input has proven instrumental in enhancing the clarity and coherence of our manuscript